# Comparison of Machine Learning Methods for Estimating Leaf Area Index and Aboveground Biomass of *Cinnamomum camphora* Based on UAV Multispectral Remote Sensing Data

Qian Wang [1], Xianghui Lu [1,*], Haina Zhang [1], Baocheng Yang [1], Rongxin Gong [1], Jie Zhang [1], Zhinong Jin [1], Rongxiu Xie [1], Jinwen Xia [1] and Jianmin Zhao [1,2]

1    Jiangxi Provincial Engineering Research Center of Seed-Breeding and Utilization of Camphor Trees, Nanchang Institute of Technology, Nanchang 330099, China
2    Jiangxi Provincial Technology Innovation Center for Ecological Water Engineering in Poyang Lake Basin, Nanchang 330029, China
*    Correspondence: xianghuilu@nit.edu.cn

**Abstract:** UAV multispectral technology is used to obtain leaf area index (LAI) and aboveground biomass (AGB) information on *Cinnamomum camphora* (*C. camphora*) and to diagnose the growth condition of *Cinnamomum camphora* dwarf forests in a timely and rapid manner, which helps improve the precision management of *Cinnamomum camphora* dwarf forests. Multispectral remote sensing images provide large-area plant spectral information, which can provide a detailed quantitative assessment of LAI, AGB and other plant physicochemical parameters. They are very effective tools for assessing and analyzing plant health. In this study, the *Cinnamomum camphora* dwarf forest in the red soil area of south China is taken as the research object. Remote sensing images of *Cinnamomum camphora* dwarf forest canopy are obtained by the multispectral camera of an unmanned aerial vehicle (UAV). Extreme gradient boosting (XGBoost), gradient boosting decision tree (GBDT), random forest (RF), radial basis function neural network (RBFNN) and support vector regression (SVR) algorithms are used to study the correlation and estimation accuracy between the original band reflectance, spectral indices and LAI and AGB of *Cinnamomum camphora*. The results of this study showed the following: (1) The accuracy of model estimation based on RF is significantly different for different model inputs, while the other four models have small differences. (2) The accuracy of the XGBoost-based LAI model was the highest; with original band reflectance as the model input, the $R^2$ of the model test set was 0.862, and the RMSE was 0.390. (3) The accuracy of the XGBoost-based AGB model was the highest; with spectral indices as the model input, the $R^2$ of the model test set was 0.929, and the RMSE was 587.746 kg·hm$^{-2}$. (4) The XGBoost model was the best model for the LAI and AGB estimation of *Cinnamomum camphora*, which was followed by GBDT, RF, RFNN, and SVR. This research result can provide a theoretical basis for monitoring a *Cinnamomum camphora* dwarf forest based on UAV multispectral technology and a reference for rapidly estimating *Cinnamomum camphora* growth parameters.

**Keywords:** *Cinnamomum camphora*; leaf area index; aboveground biomass; multispectral; band reflectance; spectral indices

## 1. Introduction

*Cinnamomum camphora* (Linn.) Presl, a broad-leaved evergreen tree of the camphor family, is mainly distributed in subtropical and tropical regions of southern China, Korea, Japan, and Vietnam [1]. Various parts of Camphoraceae (leaves, branches, trunks, fruits, etc.) are rich in fragrance substances, which are the primary raw materials in the fields of flavor and fragrance, medicinal hygiene and food [2–4]. The *C. camphora* essential oil industry has become one of the leading industries of forestry in the southern region [5]. The leaf area index (LAI) and aboveground biomass (AGB) are important indicators to evaluate

plant growth and nutrient status, among which leaf area index mainly characterizes the canopy structure of crops and is closely related to the plant processes of photosynthesis, respiration and transpiration [6]. Biomass is a crucial parameter that determines plants' light energy use, growth trend, and yield [7]. Based on the LAI and AGB, the plant health and nutrient status can be monitored for timely water and fertilizer management. The growth condition of *C. camphora* affects the leaf oil yield, affecting the quality and efficient production of the *C. camphora* industry. Thus, rapid, non-destructive and accurate monitoring of the LAI and AGB of *C. camphora* is essential for guiding the planting and management of *C. camphora* forests.

The current method of obtaining the LAI and AGB of plants uses field sampling through manual field measurements, destructive sampling, etc., which is often time-consuming, laborious and less time-efficient, limiting its large-scale application [8]. In recent years, remote sensing technology has been widely used in agriculture. Studies have used satellite remote sensing data to estimate the LAI and biomass of various crops better. Zhao et al. used HJ-1A satellite remote sensing images to estimate rice yield in the Jiangsu region [9]. Morain et al. obtained remote sensing images through the ERTS-1 satellite to estimate winter wheat yield [10] in addition to estimated crops such as cotton [11] and soybean [12]. However, satellite remote sensing images are susceptible to atmospheric influence and limited by spatial and temporal resolution and real time, resulting in the inability to obtain high-quality satellite remote sensing images suitable for precision agriculture [13,14]. For example, Landsat series satellites have a minimum repetition period of 16 days and mainly use optical sensors to acquire remote sensing images; when the signal propagation route is affected by clouds or rainfall, the images will not be applied to the accurate monitoring of plant physiological parameters [15]. Zhao et al. estimated the aboveground biomass of alpine grassland quickly and efficiently using the RF algorithm based on MODIS and SRTM data [16]. Li et al. used Sentinel-2 MSI imagery and two ensemble algorithms effectively to estimate AGB in the Shengjin Lake wetland [17]. However, satellite remote sensing is still not able to meet the needs of small-scale field trials due to more complex processing steps and insufficient spatial resolution. Hyperspectral imagers have the advantages of a wide spectral band range, high resolution and narrow bands and have been studied to predict the physiological and biochemical parameters of crops (wheat [18], maize [19], etc.) using hyperspectral remote sensing. However, the popular application of hyperspectral imagers is limited by its extensive data, redundant information that cannot be quickly processed, and high equipment prices. Compared with satellite remote sensing and hyperspectral instruments, UAV multispectral remote sensing is relatively less costly and more flexible. Multispectral remote sensing has three and more specific spectral bands, which can contain the red-edge bands important for monitoring the information of agronomic parameters, and thus, they are of great interest to the field of quantitative remote sensing in agriculture.

Many scholars have conducted much research on monitoring crops' physiological and biochemical parameters based on UAV multispectral remote sensing. For example, Qi et al. selected eight vegetation indices based on UAV multispectral images and established simple regression and artificial neural network models for estimating the leaf area index of peanuts. The results showed optimal artificial neural network estimation results [20]. Zheng et al. evaluated thirteen models for estimating the N content of winter wheat leaves using a five-band multispectral camera with nineteen vegetation indices as model-independent variables. They found that the random forest algorithm model had the highest estimation accuracy [21]. Su et al. used a UAV with a five-band multispectral camera to acquire multispectral remote-sensing images of wheat. They applied the ratio vegetation index (RVI), normalized difference vegetation index (NDVI) and optimized soil-regulated vegetation index (OSAVI) to the monitoring of yellow wheat rust with good monitoring results [22]. Feng et al. selected model inputs such as vegetation index, canopy cover and plant height to estimate cotton yield at different periods [23]. An analysis of existing research results revealed that several variables, such as spectral bands,

spectral indices and plant heights, can be used as input variables for modeling. The study demonstrated the importance of selecting the appropriate input quantities for LAI and AGB modeling estimation. The multispectral technique can acquire multiple bands of information simultaneously, and several images of different spectral bands can be obtained. Compared with soils, water bodies and rocks, etc., green plants have distinct spectral reflectance characteristics, which are mainly determined by their chemical and morphological characteristics and closely related to vegetation health. In the visible band, its spectral reflectance is mainly influenced by pigments, most visible light is absorbed by plant leaves, and the reflectance is small; in the near-infrared region, it is mainly controlled by the internal structure of plant leaves, the reflectance is large, and the absorption is low [24]. The reflectance rises sharply between the visible and near-infrared bands, forming a "red edge", the most apparent spectral feature of the plant curve and a focus of attention for vegetation remote sensing. Spectral indices are mathematical combinations based on information from different wavelengths, mainly reflecting the difference between vegetation reflectance in the visible and near-infrared wavelengths and the soil background. They can be used to quantify the growth of vegetation under certain conditions, and they are usually used to analyze parameters such as plant water content [25], leaf area index and chlorophyll [26]. In estimating the leaf area index or biomass of plants, selecting model inputs is one of the essential steps in modelling. Most of the independent variables of the estimation models were chosen as vegetation spectral indices without discussing them in the context of the original spectral reflectance. The comparison of original spectral reflectance and spectral indices in the simulation accuracy of the same inversion model rarely appears. Therefore, the analysis of band reflectance and spectral indices as model inputs for estimation accuracy must be further explored in modeling plant physiological and biochemical index estimation.

In addition to the selection of model inputs, the selection of models is also crucial. In recent years, machine learning algorithms have been widely used in precision agriculture with their superiority to invert the physicochemical parameters of plants. Pham et al. estimated the aboveground biomass of mangroves by comparing five models based on support vector regression, random forest regression, CatBoost regression, gradient augmented regression tree and extreme gradient augmented decision tree, and the results showed that the optimal model was based on extreme gradient augmented decision tree and genetic algorithm [27]. Yuan et al. estimated SPAD values for tropical endangered tree species slopes and concluded that the random forest model had the highest prediction accuracy by comparing different algorithms [28]. Siegmann et al. compared the model quality of three algorithms, support vector regression, random forest regression, and partial least squares regression, in estimating the wheat LAI and showed that the model results based on support vector regression estimation were optimal for individual years as well as for cross-validation of the entire data set [29]. Zhang et al. used stepwise regression, random forest regression and the XGBoost regression model to estimate the aboveground biomass of maize in Jilin Province, and they proposed that the XGBoost regression model had the best estimation results [30]. It has been shown that the performance of plant LAI and AGB models based on random forest, support vector regression, gradient boosted decision tree, and polar gradient augmented decision tree estimation is better, and the quality of different models applied to the estimation of different plant physicochemical parameters varies [27,31,32]. In addition, the artificial neural network is one of the common machine learning algorithms for estimating plant leaf area index and aboveground biomass, and the radial basis function neural network is one of them. The radial basis neural network learning process converges quickly and has strong robustness. Therefore, it is necessary to acquire the multispectral remote sensing images of the *C. camphora* canopy by UAV and study the influence of different models and model input selection on the inversion accuracy of *C. camphora* growth covariates. In addition, the monitoring of the growth condition of *C. camphora* using UAV multispectral remote sensing technology has rarely been reported. In this study, the UAV six-band multispectral camera was used to obtain multispectral

images of the *C. camphora* canopy and to study the correlation between the original band reflectance, spectral indices and LAI and AGB of camphor trees and the estimation accuracy. The extreme gradient boosting (XGBoost), gradient boosting decision tree (GBDT), random forest (RF), radial basis function neural network (RBFNN) and support vector regression (SVR) estimation models are constructed, respectively. The aim is to investigate the effects of different models and model inputs on the estimation of the LAI and AGB of *C. camphora* and to find the best inversion model for the LAI and AGB of *C. camphora* dwarf forests. Through accurate estimation of the LAI and AGB of *C. camphora*, we can achieve rapid monitoring of the growth of *C. camphora* dwarf forests and provide timely guidance for forest planting and management as well as provide scientific reference to promote the development of the *C. camphora* dwarf forest industry.

## 2. Materials and Methods

### 2.1. Research Area and Test Design

The experimental field of the *C. camphora* dwarf forest is located at the biotechnology experimental base of Nanchang Engineering College in Nanchang, Jiangxi Province (28°41′33″ N, 116°1′19.37″ E) (Figure 1). It has a subtropical humid monsoon climate with an average annual rainfall of 1600 mm, an average annual temperature of about 17 °C, an extreme high temperature of 42 °C and an extreme low temperature of −10 °C, and an average annual sunshine time of 1800 h with sufficient light. The soil texture of this test area was red loam with a pH of about 5.47, organic matter content of 6.39 g·kg$^{-1}$, total nitrogen content of 0.62 g·kg$^{-1}$, total phosphorus content of 0.30 g·kg$^{-1}$, total potassium content of 13.00 g·kg$^{-1}$, alkaline nitrogen content of 47.74 mg·kg$^{-1}$, fast-acting phosphorus content of 1.49 mg·kg$^{-1}$ and fast-acting potassium content of 61.10 mg·kg$^{-1}$. The test *C. camphora* were cultivated in dwarf forest; the species was "Ganfang No.1", which was from the same asexual line cuttings. It was transplanted in March 2020, planted in March 2021, and staked at the end of September 2022, when the growth of the *C. camphora* dwarf forest was basically in the harvest state and its dry matter accumulation reached its peak, and the growth condition was representative. There was a total of 66 sample plots, each with a range of 3 m × 3 m, and the planting row spacing and plant spacing were 1 m × 1 m. The fertilization treatment in the experimental field is shown in Table 1. The nitrogen fertilizer, phosphate fertilizer, and kalium fertilizer treatments were repeated twice, while the manure fertilizer and biochar fertilizer treatments were repeated three times. No drugs were used, and the grass was pulled manually.

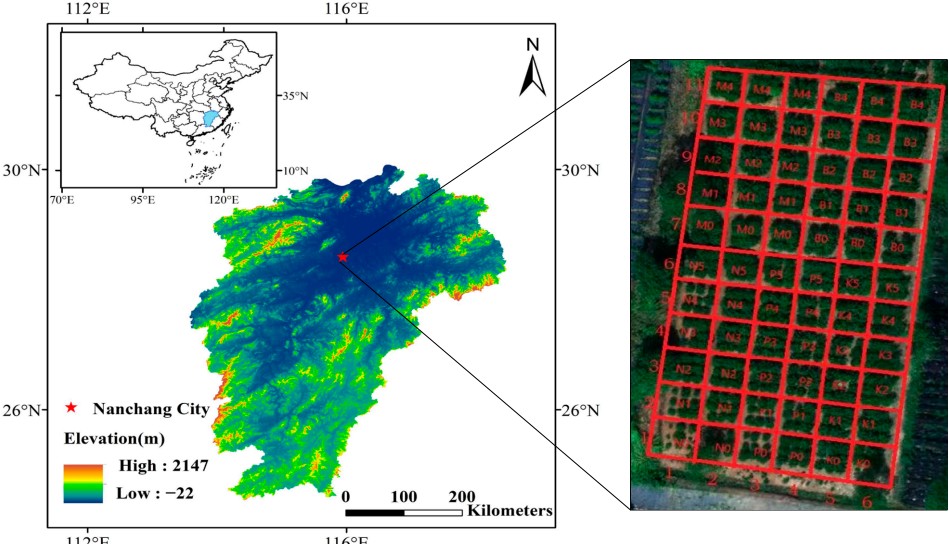

**Figure 1.** Location of the test area.

**Table 1.** Fertilization treatment of *Cinnamomum camphora*.

| Treatment | District Serial Number and Fertilization Amount (kg/hm$^2$) | | | | | |
|---|---|---|---|---|---|---|
| Nitrogen fertilizer | $N_0$ 0 | $N_1$ 45 | $N_2$ 90 | $N_3$ 135 | $N_4$ 180 | $N_5$ 225 |
| Phosphate fertilizer | $P_0$ 0 | $P_1$ 45 | $P_2$ 90 | $P_3$ 135 | $P_4$ 180 | $P_5$ 225 |
| Kalium fertilizer | $K_0$ 0 | $K_1$ 45 | $K_2$ 90 | $K_3$ 135 | $K_4$ 180 | $K_5$ 225 |
| Manure fertilizer | $M_0$ 90 | $M_1$ 67.5 | $M_2$ 45 | $M_3$ 22.5 | $M_4$ 0 | |
| Biochar fertilizer | $B_0$ 0 | $B_1$ 2500 | $B_2$ 5000 | $B_3$ 7500 | $B_4$ 10000 | |

*2.2. Data Collection*

2.2.1. UAV Remote Sensing Image Acquisition

Select a clear and windless day (26 September 2022) at approximately 12:00 p.m. to obtain the UAV remote sensing images, which reduces the handling of shadowing. The DJIM300RTK quadrotor UAV is equipped with a MS600 PRO multispectral camera, which integrates six multispectral sensor channels: blue light (central wavelength 450 nm, bandwidth 35 nm), green light (central wavelength 555 nm, bandwidth 25 nm), red light (central wavelength 660 nm, bandwidth 20 nm), red edge light 1 (central wavelength 720 nm, bandwidth 10 nm), red edge light 2 (central wavelength 750 nm, bandwidth 15 nm) and near-infrared light (central wavelength 840 nm, bandwidth 35 nm). The UAV is equipped with a six-way positioning and obstacle avoidance as well as visual flight aid interface. The multispectral camera includes "double red-edge" vegetation-sensitive bands to effectively enhance the spectral characteristics and morphological features of vegetation.

In general, the lower the flight altitude, the smaller the ground sampling distance, the higher the resolution of ground objects, and the clearer they can be seen, but the total flight time will be extended. Therefore, it is necessary to choose the optimal flight altitude according to the subject and purpose of the shooting. Considering the effect of UAV flight altitude on the accuracy of crop LAI inversion, Liu Tao et al. [33] set different flight altitudes (30, 60 and 120 m) to estimate the wheat LAI using a multispectral camera, and the results showed that the optimal estimation of LAI was obtained at an altitude of 30 m.

Based on previous inversion studies of crop parameters, this experiment sets the flight altitude to 30 m under the conditions of ensuring flight safety, general absence of obstacles, ensuring image clarity, and the UAV wind field not disturbing the crops. We set the flight path according to the scope of the test area and made whiteboard corrections, set the speed to 2.5 m·s$^{-1}$, chose the automatic capture mode, and the overlaps of the forward and side direction were both 75%. The resolution of the corresponding multispectral camera was 2.04 cm, and a total of 2766 images were captured (Table 2).

**Table 2.** The UAV system parameters.

| Parameters | Numerical and Descriptive | Parameters | Numerical and Descriptive |
|---|---|---|---|
| UAV models | DJIM300RTK | Infrared sensing of obstacle range/m | 0.1–8 m |
| Flight height | 30 m | Image sensors | 6 × 1/3″ CMOS; 1.2 million effective pixels |
| Forward overlapping | 75% | Photograph resolution/pixels | 4000 × 3000 (4:3) |
| Forward overlapping | 75% | Spatial resolution/cm | 2.04 cm |

2.2.2. Measurement of LAI and AGB

The *C. camphora* LAI was measured using the LAI-2200C Plant Canopy Analyzer (LI-COR, Lincoln, NE, USA). Compared with direct measurement methods, canopy data

measurement using a LAI-2200C Plant Canopy Analyzer can avoid the damage to vegetation caused by direct measurement methods. Moreover, the measurement operation using the instrument has the advantages of being convenient and not limited by time. It is very suitable for modern agricultural research. Three representative *C. camphora* were selected from each sample plot. For each tree, four directions (east–west, north–south) were selected to measure and record the LAI value, and the average of the four measurements was taken as the LAI value of the tree. Finally, the average of three trees was taken as the LAI value of the plot. After bringing all the *C. camphora* in each sample plot back to the laboratory, the samples were first placed in the oven to kill at 105 °C for 30 min, then dried at 80 °C for more than 48 h until constant weight, and finally, the sample mass was weighed, and the mean value was taken for each plot to measure the aboveground biomass. The ground measurement time is adopted as the same step date as the remote sensing image imaging time.

The data of 66 *C. camphora* LAI and AGB samples were obtained in this experiment, respectively. The maximum, minimum, mean, standard deviation and coefficient of variation of the LAI were 4.29, 1.26, 2.91, 0.61, and 0.21, respectively. The maximum, minimum, mean, standard deviation and coefficient of variation of AGB were 8720.09 kg·hm$^{-2}$, 2095.28 kg·hm$^{-2}$, 6398.92 kg·hm$^{-2}$, 1.63, and 0.25, respectively.

### 2.3. Data Processing

#### 2.3.1. Multispectral Data Processing

In this paper, Yusense MapV2.2.3, which is custom data processing software of the MS600 PRO multispectral camera, was used to process remote sensing images of the *C. camphora* dwarf forest canopy. Image band calibration, stitching composition, and radiation correction are performed by Yusense Map. In ENVI5.3 software, the region of interest function was used for region of interest interception. Through the steps of extracting canopy information and masking removal of shadow and soil background, the average reflectance of *C. camphora* dwarf forest canopy pixel points in the area of interest under six bands can be finally obtained separately, and the original band reflectance of *C. camphora* canopy in this sample plot can be obtained.

#### 2.3.2. Construction and Selection of Spectral Indices

Spectral indices are linear or non-linear combinations of the original band reflectance. The multispectral camera selected for this experiment has a total of six bands. In order to ensure the consistency of the number of model inputs, a total of six spectral indices were selected as inputs to the model in this experiment. The details are described below.

According to the available literature [34], the Difference Spectral Index (DSI) between any two bands is calculated separately, and the formula is shown in (1):

$$DSI = R_i - R_j \tag{1}$$

where $R_i$ and $R_j$ represent the band reflectance of blue, green, red, red edge 1, red edge 2 and any two different bands in the near-infrared band, respectively. The above spectral indices were correlated with the *C. camphora* LAI and AGB, and finally, three differential spectral indices were selected as $DSI_{NIR,R}$ (near-infrared light–red light), $DSI_{RE1,B}$ (red edge light1–blue light), and $DSI_{RE2,E}$ (red edge light2–blue light). DSI is the difference between the reflectance of two bands, which is mainly used to distinguish vegetation from soil background information [35]. In addition, in this paper, three empirical vegetation indices were selected by correlating a large number of spectral indices with the *C. camphora* LAI and AGB in a comprehensive comparative analysis, namely soil-regulated vegetation index (SAVI) [36], triangular vegetation index (TVI) [26], and overgrown vegetation index (EXG) [37].

### 2.3.3. Data Analysis and Model Accuracy Evaluation

Based on the Python 3.7 software platform, the correlation coefficients between the original band reflectance and the *C. camphora* LAI and AGB were displayed as a heatmap by the heatmap function in the seaborn library, respectively (Figure 2). The correlation coefficients between the spectral indices and the *C. camphora* LAI and AGB were displayed as a heatmap by the heatmap function in the seaborn library, respectively (Figure 3). Building the *C. camphora* LAI, the AGB estimation model was based on XGBoost and GBDT, using the Python 3.7 software platform. The RF, RBF, and SVR-based models for LAI and AGB estimation of *C. camphora* were developed using the Matlab2020a software platform. Origin2021 software was used to plot the fit of each estimation model.

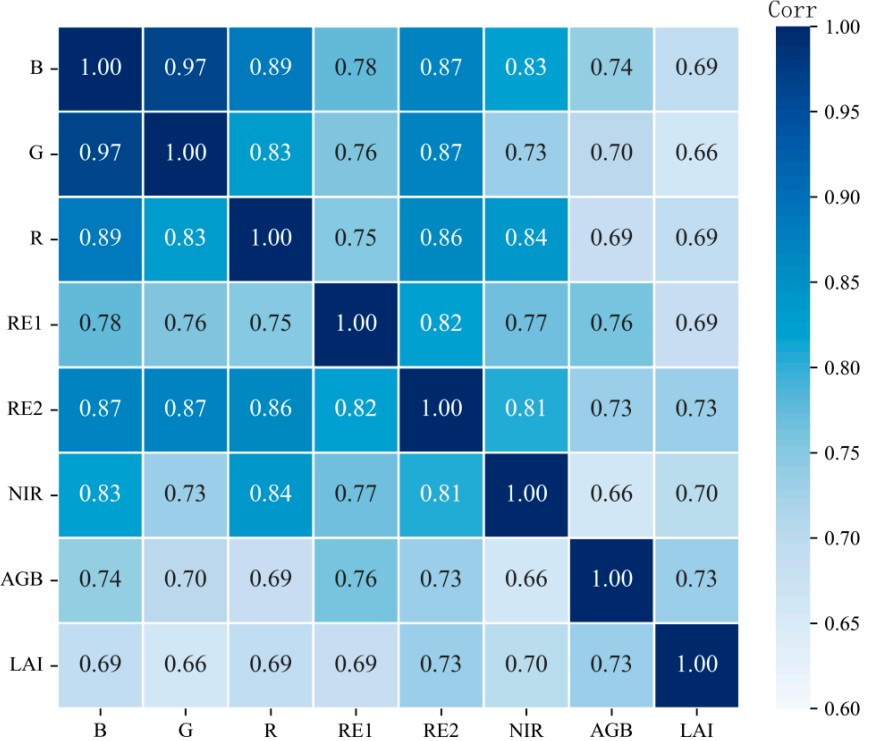

**Figure 2.** Correlation of band reflectance with *Cinnamomum camphora* LAI and AGB.

A total of 66 sets of sample data were randomly divided into 50 sets as the model training set and the remaining 16 sets as the test set. The coefficient of determination ($R^2$) and root mean square error (RMSE) were used to assess the predictive power of different algorithmic models:

$$R^2 = \frac{\sum\limits_{i=1}^{n}(\hat{y}_i - \overline{y})^2}{\sum\limits_{i=1}^{n}(y_i - \overline{y})^2} \tag{2}$$

$$RMSE = \sqrt{\frac{1}{n}\sum\limits_{i=1}^{n}(\hat{y}_i - \overline{y})^2} \tag{3}$$

where the measured value of the *C. camphora* LAI or AGB is the model estimated value of the *C. camphora* LAI or AGB, and it is the arithmetic mean of all measured values of the *C. camphora* LAI or AGB. The higher the $R^2$ value, the smaller the RMSE value, indicating a better model estimation performance.

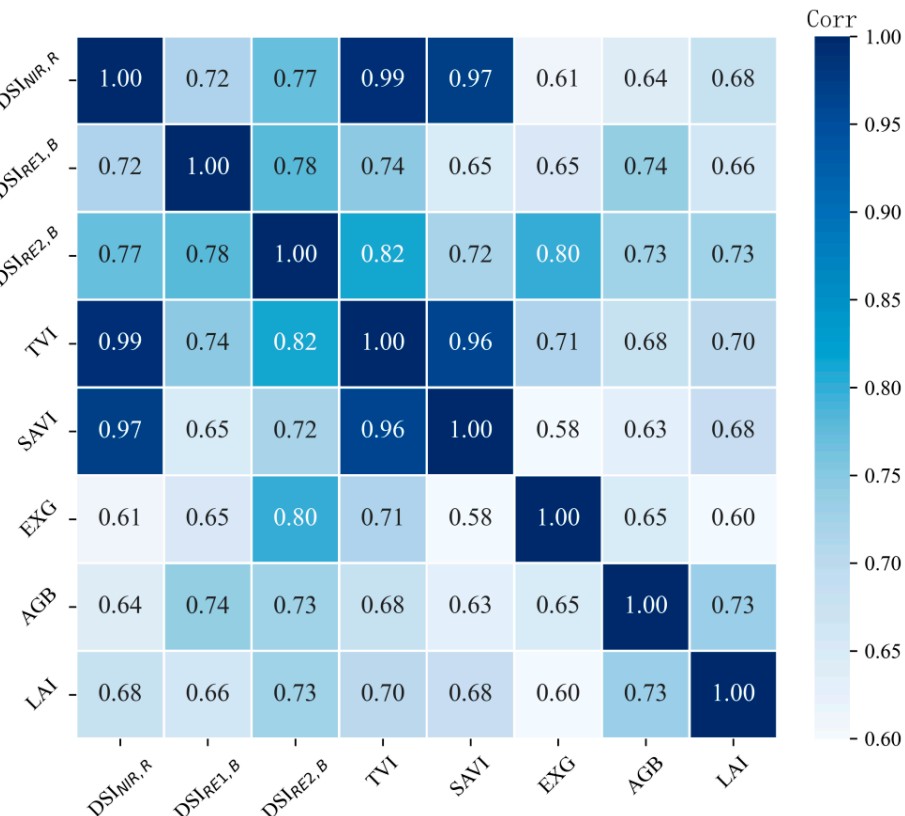

**Figure 3.** Correlation of selected spectral indices with *Cinnamomum camphora* LAI and AGB.

## 3. Results

### 3.1. Correlation Analysis of Original Band Reflectance with LAI and AGB of C. camphora

The raw band reflectance of the *C. camphora* canopy of 66 sample plots was obtained by processing and analyzing the multispectral images. A data set was formed with the corresponding 66 *C. camphora* LAI and AGB, respectively, and correlation analysis was performed. The results showed that the reflectance of blue, green, red, red edge 1, red edge 2 and NIR bands were positively correlated with the LAI and AGB (Figure 2), and the correlation coefficients were above 0.6. Among them, the correlation coefficient between the red edge 2 band reflectance and LAI is the highest at 0.731. The correlation coefficient between the green light band reflectance and LAI is the lowest at 0.659. The results sorted by correlation coefficient value from high to low were as follows: RE2 > NIR > B > R > RE1 > G.

The correlation coefficient between red edge 1 band reflectance and AGB is the highest at 0.761. The correlation coefficient between NIR band reflectance and AGB is the lowest at 0.664. The results sorted by correlation coefficient value from high to low were as follows: RE1 > B > RE2 > G > R > NIR. In addition, the leaf area index of *C. camphora* was significantly correlated with aboveground biomass with a correlation coefficient of 0.731. The correlation between the leaf area index of *C. camphora* and the original band reflectance was generally lower than that between AGB and the original band reflectance. The above analysis shows that the original band reflectance has a significant and stable correlation with the *C. camphora* LAI and AGB.

### 3.2. Correlation Analysis of Spectral Indices with LAI and AGB

Correlations between six selected spectral indices and the *C. camphora* LAI and AGB were analyzed by plotting a heatmap. The results showed that $DSI_{NIR,R}$, $DSI_{RE1,B}$, $DSI_{RE2,B}$, TVI, SAVI, and EXG significantly and positively correlated with the LAI and AGB (Figure 3), and the correlation coefficients were all above 0.6. Among them, the correlation coefficient between the $DSI_{RE2,B}$ and LAI is the highest at 0.728. The correlation coefficient between

EXG and the LAI is the lowest at 0.603. The results sorted by correlation coefficient from high to low were as follows: $DSI_{RE2,B}$ > TVI > SAVI > $DSI_{NIR,R}$ > $DSI_{RE1,B}$ > EXG.

The correlation coefficient between $DSI_{RE1,B}$ and AGB is the highest at 0.737. The correlation coefficient between SAVI and AGB is the lowest at 0.626. The results sorted by correlation coefficient from high to low were as follows: $DSI_{RE1,B}$ > $DSI_{RE2,B}$ > TVI > EXG > $DSI_{NIR,R}$ > SAVI. The above shows that the correlation levels of the *C. camphora* LAI and AGB with spectral indices are not very different, respectively. The above analysis shows that the selected spectral indices have a significant and stable correlation with the *C. camphora* LAI and AGB.

### 3.3. Construction of LAI and AGB Estimation Model Based on Band Reflectance and Spectral Indices

From Figures 2 and 3, it can be seen that this paper can use six original band reflectance and six spectral indices as the input quantities of the *C. camphora* LAI and AGB inversion models, respectively. Modeling using XGBoost, GBDT, RF, RBFNN and SVR algorithms, respectively, the results based on the XGBoost model are shown in Figure 4. The model prediction accuracy comparison results are shown in Table 1.

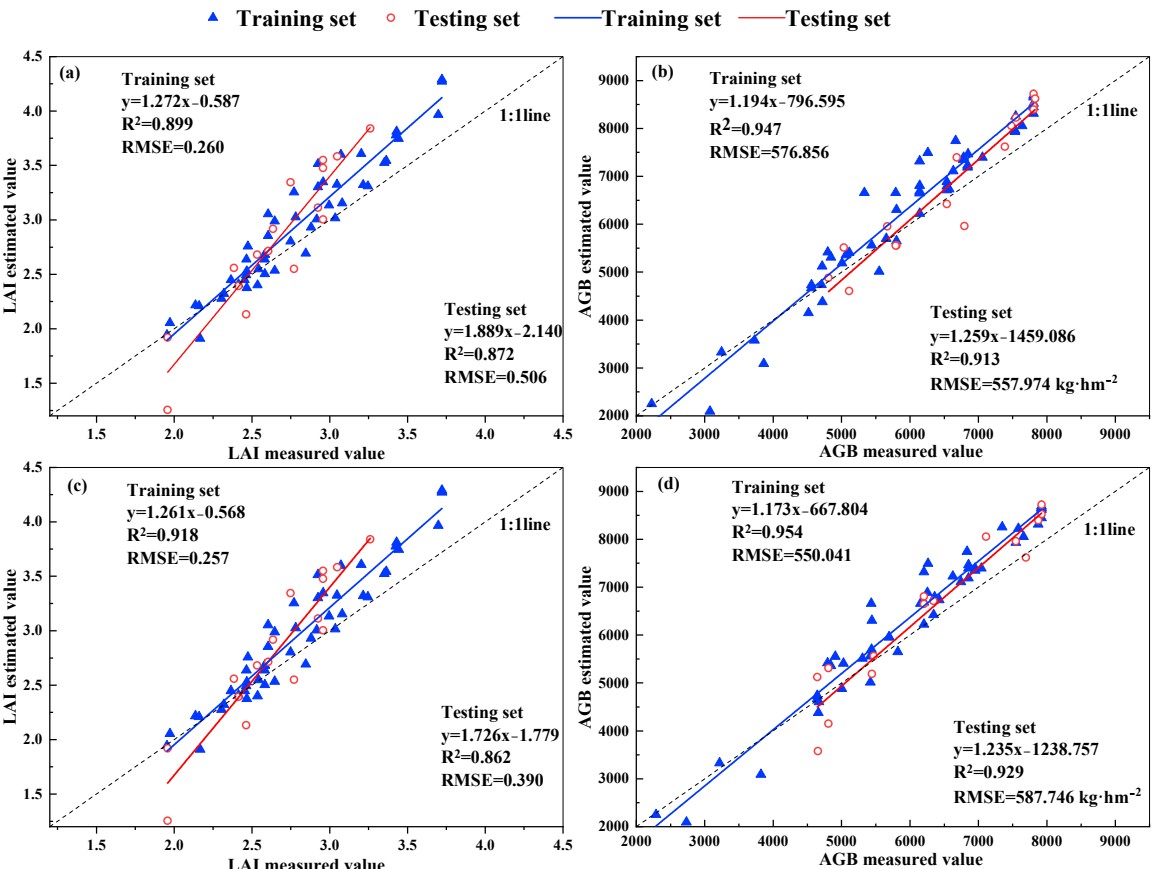

**Figure 4.** Prediction results of the training and test sets of the XGBoost-based *Cinnamomum camphora* LAI and AGB estimation models. (**a**) The estimated LAI input variable is band reflectance; (**b**) the estimated AGB input variable is band reflectance; (**c**) the estimated LAI input variable is spectral indices; (**d**) the estimated AGB input variable is the spectral indices.

This paper uses network search and five-fold cross-validation to tune the model parameters of XGBoost, GBDT and RF. The RBFNN and SVR models are manually tuned using a for-loop and cross-validation strategy. Based on the XGBoost algorithm to estimate the *C. camphora* LAI and AGB, the optimal parameters are refined by a grid search to find the best parameters and then delineate more detailed intervals. In the final model parameters, the number of weak learners (n_estimators) are set to 100, the shrinkage step

(learning_rate) is set to 0.03, 0.02, and the maximum depth of the decision tree (max_depth) is set to 5. Based on the GBDT algorithm to estimate the *C. camphora* LAI and AGB, the model parameters n_estimators were set to 100, and the learning_rates were set to 0.03 and 0.02, respectively, while the max_depth was set to 3. Based on the RF algorithm to estimate the LAI and AGB, the model parameter trees were set to 100, and the minimum number of leaves was set to 1 and 2, respectively. For the estimation of the *C. camphora* LAI and AGB based on RBFNN, with LAI spectral indices and band reflectance as independent variables and the expansion rate of radial basis function (rbf_spread) set to 90 and 120, respectively, with AGB spectral indices and band reflectance as independent variables, the expansion rate of the radial basis function was set to 50 and 70, respectively. For the estimation of the *C. camphora* LAI and AGB based on SVR, with LAI spectral indices and band reflectance as independent variables, penalty factor (C) values were set to 4 and 8, and radial basis parameters (g) were set to 0.0625 and 0.0313, respectively. The AGB spectral indices and band reflectance were used as independent variables; the penalty factor (C) values were set to 1 and 16, and the radial base parameters (g) were set to 4 and 2, respectively.

From Table 3, among the five models, the $R^2$ values of the model's training and test sets for estimating the *C. camphora* LAI and AGB are above 0.6 and up to 0.927, indicating that the model can be used for the estimation of *C. camphora* LAI and AGB. Based on the RF estimation of the LAI and AGB, the $R^2$ of the test set of the model with the spectral indices as the independent variable is higher than the band reflectance, under which the spectral indices contain more information on the spectral features related to the *C. camphora* LAI and AGB. The $R^2$ of the model test set with different input quantities of the remaining four models of the estimated *C. camphora* LAI was similar, and the RMSE values were all lower. Overall, the model estimation accuracy was similar between different models with different inputs to estimate the same index (LAI or AGB).

**Table 3.** Comparison of precision results of estimation models.

| Item | Model | Input Variable | Training Set | | Testing Set | |
|---|---|---|---|---|---|---|
| | | | $R^2$ | RMSE | $R^2$ | RMSE |
| LAI | XGBoost | spectral indices | 0.918 | 0.257 | 0.862 | 0.390 |
| | | band reflectance | 0.899 | 0.260 | 0.872 | 0.506 |
| | GBDT | spectral indices | 0.922 | 0.193 | 0.856 | 0.347 |
| | | band reflectance | 0.915 | 0.265 | 0.852 | 0.264 |
| | RF | spectral indices | 0.918 | 0.191 | 0.855 | 0.400 |
| | | band reflectance | 0.927 | 0.184 | 0.823 | 0.445 |
| | RBFNN | spectral indices | 0.724 | 0.331 | 0.701 | 0.695 |
| | | band reflectance | 0.778 | 0.270 | 0.716 | 0.522 |
| | SVR | spectral indices | 0.621 | 0.375 | 0.619 | 0.440 |
| | | band reflectance | 0.628 | 0.379 | 0.609 | 0.453 |
| | Model | Input Variable | $R^2$ | RMSE (kg·hm$^{-2}$) | $R^2$ | RMSE (kg·hm$^{-2}$) |
| AGB | XGBoost | spectral indices | 0.954 | 550.041 | 0.929 | 587.746 |
| | | band reflectance | 0.947 | 576.856 | 0.913 | 557.974 |
| | GBDT | spectral indices | 0.917 | 597.758 | 0.894 | 509.739 |
| | | band reflectance | 0.915 | 577.032 | 0.909 | 725.561 |
| | RF | spectral indices | 0.873 | 598.332 | 0.862 | 827.374 |
| | | band reflectance | 0.873 | 628.086 | 0.845 | 667.928 |
| | RBFNN | spectral indices | 0.749 | 845.087 | 0.717 | 941.614 |
| | | band reflectance | 0.751 | 863.428 | 0.729 | 708.960 |
| | SVR | spectral indices | 0.751 | 844.610 | 0.749 | 712.076 |
| | | band reflectance | 0.763 | 801.306 | 0.741 | 801.233 |

The XGBoost model outperformed other algorithmic models in estimating the LAI and AGB of *C. camphora* with different models. The XGBoost model estimated the *C. camphora* LAI best with the band reflectance as the input quantity; the $R^2$ values of the model training

set and test set were 0.899 and 0.872, and the RMSE values were 0.260 and 0.506, respectively. The XGBoost model has the highest accuracy in estimating the AGB of *C. camphora* with the input quantity, and the $R^2$ values of the model training set and test set were 0.954 and 0.929, and the RMSE values were 550.041 kg·hm$^{-2}$ and 587.746 kg·hm$^{-2}$, respectively, indicating that the XGBoost model is the best model for estimating the LAI and AGB of *C. camphora*. In addition, the GBDT model was modeled extremely fast (within 0.037 s) under Python 3.7, which was better than the remaining four models.

## 4. Discussion

The leaf area index and aboveground biomass are essential indicators of the growth of *C. camphora* dwarf forests, and determining the sensitive spectral indices of the two parameters is the key to achieving rapid and non-destructive spectral monitoring. In this study, we analyzed the correlation between the reflectance of blue light, green light, red light, red edge 1, red edge 2, and near-infrared bands and the leaf area index and aboveground biomass of *C. camphora*. The study's results found that the leaf area index and aboveground biomass of *C. camphora* had the highest correlation with the red edge band. Fan et al. concluded that the leaf area index of oilseed rape at different fertility stages was more significantly correlated with NIR band reflectance, which is different from the present study results, indicating that different plants are sensitive to different spectral bands [38]. The red-edge band is a sensitive band to characterize the growth of green plants, which is closely related to important biochemical parameters such as plant chlorophyll, biomass and leaf area index [39]. In existing studies, a large number of spectral indices have been applied to quantitative remote sensing in agriculture to monitor plant growth, such as wheat [40], peanut [20] and maize [41]. The different spectral indices selected in this paper were significantly correlated with the leaf area index and biomass of *C. camphora*, especially the spectral indices with the addition of red-edge band reflectance. Dong et al. studied the effect of red-edge reflectance-based and visible reflectance-based vegetation indices for the leaf area index estimation of spring wheat and oilseed rape, and they found that red-edge reflectance-based spectral indices can be used to develop more general LAI estimation models for different crops, similar to the results of this study [42]. Mutanga et al. predicted biomass in vegetation-intensive wetland areas based on narrow-band spectral indices calculated in the red-edge and near-infrared bands, effectively improving estimation accuracy [43]. The result is that the red-edge band reflectance and the spectral indices based on the red-edge band reflectance strongly correlate with the *C. camphora* LAI and AGB.

Among the five models used in this study, the $R^2$ of the model test set for estimating *C. camphora* AGB was higher than that of the LAI, indicating that the accuracy of the *C. camphora* AGB estimation model was generally higher than that of the LAI estimation model under the same modeling approach, probably because the aboveground biomass was more intuitive, which is consistent with the findings of Bascon et al. [44]. The RMSE values of the estimated *C. camphora* AGB models were all much greater than those of the LAI because the RMSE values were more sensitive to the magnitude, and the AGB values were of a larger order of magnitude. The models' RMSE values are smaller than the *C. camphora* AGB sample data range 2096 to 8721.

In this paper, through correlation analysis, six spectral indices (DSI$_{RE2,B}$, TVI, SAVI, DSI$_{NIR,R}$, DSI$_{RE1,B}$, EXG) and six band reflectance values are selected as model-independent variables. Based on GBDT, RF, and SVR estimation models for the LAI, using spectral indices as model-independent variables can better estimate the *C. camphora* LAI; based on XGBoost and RBFNN estimation models, the accuracy of the LAI estimation is better with the band reflectance as the model input. For *C. camphora* AGB, based on XGBoost, GBDT, RF, and SVR estimation models, the spectral indices as the model-independent variable can better estimate *C. camphora* AGB; based on the RFNN estimation model, the better model independent variable is the band reflectance. In the same model, the best independent variable for LAI and AGB estimation was the spectral indices, which was probably because the spectral indices are a linear or non-linear combination of multiple

band reflectance, which can better characterize the spectral information of the LAI and AGB of *C. camphora*. In addition, based on the RF estimation models, the differences between different independent variables were more apparent. The remaining four models had differences but were few, indicating that the variable selection had different effects on the model estimation accuracy.

To estimate the leaf area index of *C. camphora*, the RFNN model outperforms the SVR model; to estimate the aboveground biomass of *C. camphora*, the SVR model outperforms the RFNN model, and the $R^2$ of the two model test sets ranges from 0.60 to 0.75. The accuracy of SVR to estimate *C. camphora* AGB is higher than that of the LAI, but the $R^2$ of the model is still lower, which may be the limitation of the model kernel function and penalty factor C and other parameters themselves [40]. The reason for the low accuracy of the RFNN estimation *C. camphora* AGB model may be that the model has too many initial values central in the LAI or AGB input sample set, and the central values of the selected hidden layer basis functions are complex to reflect the actual output relationship of the system in most cases [45]. The $R^2$ values of the RF, GBDT, and XGBoost-based models for estimating the leaf area index and aboveground biomass of *C. camphora* were above 0.8, which indicated the excellent estimation ability of the three models for estimation. The accuracy of RF, GBDT, and XGBoost-based AGB estimation models for *C. camphora* is generally higher than that of LAI estimation models, and all three algorithms belong to decision tree-based machine learning algorithms, which have been shown to have more obvious advantages in estimating plant biomass [31,32]. It was found that the RF-based model is second only to XGBoost and GBDT in estimation accuracy and also has a better estimation effect ($R^2$ of 0.82 or more). RF is not easy to overfit, has good resistance to noise and has the advantages of being able to perform variable importance calculation and ranking due to its few adjustable parameters, including the speed, efficiency, and ability to perform variable importance calculation and ranking. However, its complexity makes the training time longer than other similar algorithms [46]. The GBDT-based model estimation accuracy is slightly second to XGBoost, with a model $R^2$ of 0.85 or more, and the performance of GBDT is a step up from RF, which is more suitable for the low-dimensional data used in this paper and has a higher estimation accuracy with relatively less tuning time [47]. The XGBoost-based model had the highest accuracy in estimating the leaf area index and aboveground biomass of *C. camphora* (Table 1), which was better than other algorithmic models, which is consistent with the findings of Zhang et al. [48], who used hyperspectral images of drones to estimate the leaf area index of winter wheat, and Li et al. [49], who used satellite images to estimate the forest biomass of different types in the Xiangjiang River basin in Hunan Province. The estimation accuracy of GBDT-based models is slightly lower than that of XGBoost, which is probably because XGBoost further optimizes the loss function for GBDT, and XGBoost can automatically employ CPU multi-threading for parallel computation, simplifying the model while improving prediction accuracy [50]. The XGBoost algorithm is currently the fastest open-source boosting tree toolkit with good processing speed and accuracy for low and medium-dimensional data. Researchers have widely used it in many fields [51–54], but its potential has yet to be fully exploited in forestry.

In addition, "Ganfang No. 1" planted in this experiment is an excellent variety of spice *C. camphora* declared by the Jiangxi Camphor Breeding and Development and Utilization Engineering Research Center, which performs well in crucial indices such as oil yield and linalool content. The findings of this study cannot be directly applied in different spatial and temporal *C. camphora* dwarf forest plantations because the LAI and AGB may reflect different spectral characteristics in multispectral remote sensing images under the influence of different varieties, different times and the same geographical conditions. Therefore, the modeling estimation can be carried out separately under different conditions in the subsequent study to improve the broad applicability of the model.

## 5. Conclusions

In this study, the correlation coefficients between the spectral indices, band reflectance and leaf area index and biomass of *C. camphora* were calculated separately using Pearson correlation analysis, and the results showed that the correlations were significant. Among the selected band reflectance and spectral indices, the *C. camphora* LAI had the highest correlation with red edge 2 band reflectance and $DSI_{RE2,B}$, respectively, and the *C. camphora* AGB had the highest correlation with red edge 1 band reflectance and $DSI_{RE1,B}$, respectively. The spectral indices based on the red edge band correlated better with the LAI and AGB. Five algorithms were used to estimate the *C. camphora* leaf area index and aboveground biomass using spectral indices and the original band reflectance as model inputs, respectively. Through a comprehensive comparative analysis, there were more apparent differences in the accuracy of the RF-based estimation models with different model inputs. The remaining four models had differences, but there were few. Under the same modeling method, the accuracy of the AGB estimation model was generally higher than that of the LAI estimation model. After testing different modeling methods, the XGBoost model estimation accuracy was the highest, with the model test set $R^2$ value of 0.862 and RMSE value of 0.390 using the band reflectance as the model input, and the model test set $R^2$ value of 0.929 and RMSE value of 587.746 kg·hm$^{-2}$ using the spectral indices as the model input. The study showed that based on the UAV, the estimation of LAI and AGB of *C. camphora* dwarf forest based on multispectral remote sensing images by XGBoost has a good prediction effect, which is a guide for the management of *C. camphora* dwarf forest.

**Author Contributions:** Conceptualization, J.Z. (Jie Zhang) and Z.J.; methodology, R.G. and B.Y.; software, Q.W.; validation, H.Z., B.Y. and Q.W.; formal analysis, Q.W.; investigation, Q.W. and B.Y.; resources, J.Z. (Jie Zhang) and X.L.; data curation, Q.W. and R.X.; writing—original draft preparation, Q.W.; writing—review and editing, X.L.; visualization, J.X.; supervision, J.Z. (Jianmin Zhao); project administration, X.L.; funding acquisition, X.L. and J.Z. (Jie Zhang). All authors have read and agreed to the published version of the manuscript.

**Funding:** This research was funded by the National Natural Science Foundation of China, grant number 52269013 and 32060333, the Natural Science Foundation Project of Jiangxi Province grant number 20232BAB205031, Jiangxi province main discipline academic and technical leaders training plan youth project of China, grant number 20204BCJL23046, Jiangxi Provincial Science and Technology Department Major Science and Technology Project of China, grant number 20203ABC28W016-01-04, Jiangxi Forestry Bureau camphor tree research project of China, grant number 202007-01-04.

**Data Availability Statement:** Not applicable.

**Acknowledgments:** We wish to thank Youzhen Xiang and her team for the experimental site. We wish to thank Jie Zhang and her team for the experimental site.

**Conflicts of Interest:** The authors declare no conflict of interest.

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
