# Peer review of "Comparison of Machine Learning Methods for Estimating Leaf Area Index and Aboveground Biomass of Cinnamomum camphora Based on UAV Multispectral Remote Sensing Data"

_forests, doi:10.3390/f14081688_

Round 1

Reviewer 1 Report

The article describes the methods of estimation LAI and AGB based on UAV data. The authors use multispectral UAV data and compare several machine learning methods. The problem is relevant, but the manuscript needs to be improved.

1. The Introduction doesn't say anything about the global products of the LAI and AGB that MODIS, ESA, etc. regularly release. They usually contain averaged values over a period, which reduces the influence of weather and atmosphere that you refer to when describing satellite data.

22.       Why were these three spectral and three vegetation indices chosen? Justify this choice. (Subsection “Construction and selection of spectral index”)

33.       You give results of machine learning models separately for spectral indices and band reflections. Why didn't you use these data for training together, bands+indices? Often, increasing the amount of input data for training improves its accuracy.

44.       The numbering of sections and subsections is incorrect, from 1.2 “Data collection” (line 180) onwards.

55.       The paper should be carefully proofread. So far, I have noticed mistakes and unclear sentences that make your research worse. For example: “Import the tif file into ENVI”, “Mapping the correlation heat map”, etc. - Is it an instruction, or your processing step, or a subheading?

 The English should be polished. For example, Line 269, the sentence “The highest correlation between the red-edge 1-band…” does not contain a predicate.

Author Response

Dear reviewer,

We sincerely thank the reviewer for carefully examining our manuscript and providing us with helpful comments to assist in the revision. We would like to express our sincere gratitude to you for the constructive comments and suggestions. Our responses to the comments are provide below.

  1. The Introduction doesn't say anything about the global products of the LAI and AGB that MODIS, ESA, etc. regularly release. They usually contain averaged values over a period, which reduces the influence of weather and atmosphere that you refer to when describing satellite data.

Response:

We have revised the manuscript to address your concerns and hope that it is now clearer. Please see page 2 of the revised manuscript, lines 67–72. Thanks for your correction.

  1. Why were these three spectral and three vegetation indices chosen? Justify this choice. (Subsection “Construction and selection of spectral index”)

Response:

(1) Eight commonly used vegetation indices were calculated to estimate rice LAI using 12-band UAV images by Gong et al. (https://doi.org/10.1186/s13007-021-00789-4) Tang et al. estimated LAI and AGB of winter oilseed rape by UAV multispectral, selected 14 spectral indices (containing vegetation indices) and divided them into three groups as input variables of the model, and obtained a better estimation effect by using Random Forest algorithm.(https://doi.org/10.3390/agronomy12071729). Based on existing studies, it is feasible to use spectral indices, which have high correlation with crop physiological and biochemical indicators, as model inputs. The commonly used vegetation indices and spectral indices can be applied in combination with each other to complement each other in vegetation monitoring and research, and they are strongly correlated. The combination of the vegetation indices and spectral indices has the potential to provide more comprehensive information on vegetation.

(2) This paper focuses on the impact of different models and the selection of model inputs on the estimation results. The multispectral camera that we chose six bands of reflectance. To ensure the reliability of the test as much as possible, we kept the number of model inputs constant. Therefore, we analyzed the correlation of a large number of vegetation indices and spectral indices with LAI and AGB, and selected the three spectral indices and three vegetation indices with the highest correlation as model independent variables in a comprehensive way.

(3) After reading a large amount of literature on crop parametric inversion, we found that the choice of spectral indices (number, type) is not clearly and strictly defined, and is mostly based on the characteristics and actual situation of the study crop itself. In summary, we have added to our resubmitted manuscript. Please see page 7 of the revised manuscript, lines 253–256.

  1. You give results of machine learning models separately for spectral indices and band reflections. Why didn't you use these data for training together, bands+indices? Often, increasing the amount of input data for training improves its accuracy.

Response:

(1) Using both spectral indices and band reflectance as model inputs essentially does not increase the total number of samples, but only the number of independent variables. To some extent, more independent variables will improve the estimation accuracy. However, too many independent variables can also reduce the efficiency of the method.

(2) The two different inputs (spectral indices and band reflectance) have their own properties, which leads to different ways of mapping them to the same output. Different types of samples are fed into the model for training, which tends to confuse the model and often gives poor results. In the presence of information redundancy between independent variables, there is little improvement in estimation accuracy.

  1. The numbering of sections and subsections is incorrect, from 1.2 “Data collection” (line 180) onwards.

Response:

We are sorry for our carelessness. In our resubmitted manuscript, the numbering of sections and subsections are revised. Changes in numbering have been highlighted in yellow. Thanks for your correction.

  1. The paper should be carefully proofread. So far, I have noticed mistakes and unclear sentences that make your research worse. For example: “Import the tif file into ENVI”, “Mapping the correlation heat map”, etc. - Is it an instruction, or your processing step, or a subheading?

Response:

We have carefully reviewed the manuscript and made changes accordingly, including partial correction of typos, grammatical errors and long sentences. Please see page 6 of the revised manuscript, lines 245–248 and page7, lines 295 and other sections highlighted in yellow. Thanks for your constructive suggestions.

  1. The English should be polished. For example, Line 269, the sentence “The highest correlation between the red-edge 1-band…” does not contain a predicate.

Response:

We had thoroughly reviewed the paper for clarity and modified the language for native English speakers. And here we did not list the changes but marked in yellow in the revised paper.

We would like to thank the referee again for taking the time to review our manuscript. Looking forward to hearing from you.

Your sincerely,

Corresponding author: Xianghui Lu

Reviewer 2 Report

The topic selection and research of the paper have certain significance, but there are the following issues that need to be addressed.

a)      What is the impact of flight altitude and resolution on the experimental results? It is necessary to describe.

b)      The impact of weather on multispectral data collection, such as clouds, fog, etc

c)      What is the specific distribution form of the training set and the verification set in this paper, and is the training set and the verification set selected in the form of random distribution ?

d)      In this paper, a total of 66 samples were selected. Is there a slight shortage in the number of samples.  The ratio of training set and testing set is generally 8:2 or 7:3

e)      In line 241, how does this paper relate C.camphora LAI, AGB to the original band reflectance and spectral index ?

f)       the Materials and Methods section needs to provide more of an explanation as to what method did you use and why, you need to strength them.

g)      the mean square error should have units.

h)      the secondary heading number of Part 2 is mislabeled.

Minor editing of English language required

Author Response

Dear reviewer,

We sincerely thank the reviewer for carefully examining our manuscript and providing us with helpful comments to assist in the revision. We would like to express our sincere gratitude to you for the constructive comments and suggestions. Our responses to the comments are provide below.

  1. a) What is the impact of flight altitude and resolution on the experimental results? It is necessary to describe.

Response:

In general, the lower the flight altitude, the smaller the ground sampling distance, the higher the resolution of ground objects, the clearer they can be seen, but the total flight time will be extended. Therefore, it is necessary to choose the optimal flight altitude according to the subject and purpose of the shooting. Please see page 5 of the revised manuscript, lines 200–212. Thanks for your constructive suggestions.

  1. b) The impact of weather on multispectral data collection, such as clouds, fog, etc

Response:

(1) If the UAV goes through the clouds, these small droplets of water are likely to penetrate the fuselage and cause some of the electronics to short-circuit. However, the UAV flight altitude for this test was set at 30 m, which is far away from the cloud, making the cloud hazard factor negligible. Therefore, the cloud had no effect on the UAV multispectral data acquisition for collecting multispectral data.

(2) When flying in fog, the surface of the UAV can become very wet, potentially leading to water ingress into the internal components of the drone body and damage to the drone. At the same time, the atomized lens is not clear enough to take a picture, which makes the task of multispectral data acquisition impossible.

(3) Flying under cloudy skies, with slightly higher air humidity than on clear days and poor lighting, reduces the amount of information in raw multispectral images.

(4) Flying in rainy weather potentially poses a risk of short-circuiting the drone.

(5) The safety of the test cannot be guaranteed in the case of high winds, which may cause the machine to fall out of control due to the loss of control of the drone.

In summary, weather such as fog, rain, and high winds can potentially lead to UAV damage, poor multispectral data quality, and so on. In this paper, a clear and windless day (September 26, 2022) was chosen to acquire the UAV remote sensing images at approximately 12 noon. In addition, this reduces the handling of shadows. We have made additions to the revised manuscript. Please see page 5 of the revised manuscript, lines 189–190. Thanks for your correction.

  1. c) What is the specific distribution form of the training set and the verification set in this paper, and is the training set and the verification set selected in the form of random distribution?

Response:

In this paper, a total of 66 sets of sample data are used, and the data are randomly divided into 50 sets as the model training set and the remaining 16 sets as the test set.

  1. d)   In this paper, a total of 66 samples were selected. Is there a slight shortage in the number of samples.  The ratio of training set and testing set is generally 8:2 or 7:3

Response:

(1)     From the multispectral images, the trees in each plot are closely connected, resulting in each tree not being able to be used as a sample point by itself. So the sample size does seem a bit small. However, to compensate for its limitations, this paper uses cross-validation to train the optimal selection model. First, the sample data are randomly divided into two parts (50 training sets and 16 test sets), then the training sets are used to train the model and the model and parameters are validated on the test sets. Next, we then disrupt the samples, re-select the training and test sets, and continue to train the data and test the model. Finally, we choose the loss function to evaluate the optimal model and parameters.

(2) For the same input quantity (original band reflectance or spectral indices), this experiment focuses on comparing the accuracy of different algorithms (XGBoost, GBDT, RF, RBFNN, SVR) for estimating the LAI and AGB. Therefore, the estimation differences of different algorithms can be effectively compared on the same level of data volume.

(3) For the traditional machine learning phase (datasets in the order of 10,000), the typical allocation ratio is 7:3 or 8:2 between the training set and the test set. However, data acquisition has some limitations due to the fact that in field trials. In this paper, a ratio of approximately 7:3 (50:16 in this paper) is used to divide the training and test sets, which slightly increases the number of training sets to more accurately reflect the efficacy of the different models, to the extent feasible.

  1. e) In line 241, how does this paper relate C.camphora LAI, AGB to the original band reflectance and spectral index ?

Response:

Based on the Python 3.7 software platform, the correlation coefficients between the original band reflectance and the C. camphora LAI and AGB were displayed as heatmap by the heatmap function in the seaborn library, respectively (Figure 2). The correlation coefficients between the spectral indices and the C. camphora LAI and AGB were displayed as heatmap by the heatmap function in the seaborn library, respectively (Figure 3). We have revised the manuscript to address your concerns and hope that it is now clearer. Please see page 7 of the revised manuscript, lines 268–272. Thanks for your correction.

  1. f) the Materials and Methods section needs to provide more of an explanation as to what method did you use and why, you need to strength them. the Materials and Methods section needs to provide more of an explanation as to what method did you use and why, you need to strength them.

Response:

We have made additional content in Materials and Methods and hope that it is now clearer. Please see page 5 of the revised manuscript, lines 200–212 and page 6, lines 221–226, lines 245–248, lines 253–256. Thanks for your constructive suggestions.

  1. g) the mean square error should have units. the mean square error should have units.

Response:

We feel sorry for our carelessness. In our resubmitted manuscript, we have added the RMSE to the corresponding unit. Please see page 10 of the revised manuscript, Table 3, lines 371 and page 11, Figure4. Thanks for your constructive suggestions.

  1. h) the secondary heading number of Part 2 is mislabeled.

Response:

In our resubmitted manuscript, the numbering of sections and subsections are revised. Changes in numbering have been highlighted in yellow. Thanks for your correction.

We had thoroughly reviewed the paper for clarity and modified the language for native English speakers. We would like to thank you again for taking the time to review our manuscript. Looking forward to hearing from you.

Your sincerely,

Corresponding author: Xianghui Lu

Round 2

Reviewer 1 Report

Good work, thank you! All my suggestions are adjusted.